# Aquaporin-5 Dynamic Regulation

**DOI:** 10.3390/ijms24031889

**Published:** 2023-01-18

**Authors:** Claudia D’Agostino, Dorian Parisis, Clara Chivasso, Maryam Hajiabbas, Muhammad Shahnawaz Soyfoo, Christine Delporte

**Affiliations:** 1Laboratory of Pathophysiological and Nutritional Biochemistry, Université Libre de Bruxelles, 1070 Brussels, Belgium; 2Rheumatology Department, CUB Hôpital Erasme, Hôpital Universitaire de Bruxelles (H.U.B), Université Libre de Bruxelles (ULB), Route de Lennik 808, 1070 Brussels, Belgium

**Keywords:** aquaporin, regulation, effectors, trafficking, function, mammalian

## Abstract

Aquaporin-5 (AQP5), belonging to the aquaporins (AQPs) family of transmembrane water channels, facilitates osmotically driven water flux across biological membranes and the movement of hydrogen peroxide and CO_2_. Various mechanisms have been shown to dynamically regulate AQP5 expression, trafficking, and function. Besides fulfilling its primary water permeability function, AQP5 has been shown to regulate downstream effectors playing roles in various cellular processes. This review provides a comprehensive overview of the current knowledge of the upstream and downstream effectors of AQP5 to gain an in-depth understanding of the physiological and pathophysiological processes involving AQP5.

## 1. Introduction

Aquaporin-5 (AQP5) belongs to the family of transmembrane aquaporins (AQPs) [1,2] and, more specifically, to the classical subfamily (including AQP0, AQP2, AQP4, AQP5, AQP6, and AQP8) characterized by high sequence homology between species and selective water permeability [3]. Indeed, the expression of rat and human *Aqp5* (*rAqp5* and *hAqp5*) in Xenopus oocytes has revealed its selective permeability to water [4,5]. In addition, hAQP5 is permeable to small molecules, including hydrogen peroxide [6] and CO_2_, but not NH_3_ [7,8], suggesting an important role in cell survival and adaptation to oxidative conditions. 

The common structural features of AQPs, including AQP5, consist of six transmembrane domains (TM 1 to 6), five connecting loops (CL A to E) with loops B and E from opposite membrane sides folding back to form shorter pseudo-transmembrane helices within opposite layers of the membrane bilayer, and intracellular N- and C- termini [9,10,11]. Four monomers, each forming a water-permeable pore, associate in a tetramer presenting an additional central pore that was proposed to be permeable to gas or hydrophobic molecules [12]. Two asparagine-proline-alanine (NPA) motifs, located within the ends of the short pseudo-transmembrane helices formed by loops B and E, and an aromatic-arginine (ar/R) region, constituting the narrowest part of the pore, have been suggested to prevent the movement of protons along with water molecules by a Grotthuss-type mechanism. The solute specificity of AQPs relies on the variable composition of the ar/R region, except for a highly conserved arginine [9,10]. 

Within the human body, *Aqp5* is expressed in various secretory tissues (such as salivary glands, pancreas, lacrimal glands, mammary glands, thyroid, adrenals, pituitary, skin), internal tissues (such as lungs, kidney, liver, spleen, esophagus, stomach, intestine, colon, bladder, adipose tissue), reproductive tissues (such as testes, ovaries, prostate, uterus), muscle tissues (such as heart, skeletal muscle, arteries), nervous tissues (such as brain, cortex, cerebellum, spinal cord, tibial nerve), immune tissues (such as lymph nodes), and ocular tissues (cornea, lens, retina) [13,14,15,16,17]. AQP5 plays an important physiological role in the production of saliva, sweat, tears, and pulmonary secretions [18,19,20,21] and optical properties of the lens [22,23,24].

AQP5, like other AQPs, encounters dynamic regulation consisting of transcriptional and translational modulation, post-translational modifications, and trafficking to the plasma membrane. Therefore, such dynamic regulation plays a key role in physiological and pathophysiological processes.

In this review, we provide an in-depth overview of the current understanding of the AQP5 dynamic regulation. 

## 2. Functional Impact of Single Nucleotide Polymorphism and Mutation of *Aqp5* Gene

Single nucleotide polymorphisms and mutations of the *Aqp5* gene have been associated with a perturbation in mRNA expression, protein trafficking, and water permeability.

The CT/TT genotype of *Aqp5* rs1964676 polymorphism has been associated with strong *Aqp5* expression and used as a prognostic marker of survival in patients with early breast cancer [25]. 

The sequencing of the *Aqp5* gene promoter region of 50 healthy Caucasian blood donors has identified a common single nucleotide polymorphism characterized by the substitution of C for A at position -1364 (-1364A/C) [26]. This genetic modification was associated with increased binding of transcription factors to the promoter but with reduced cAMP-induced *Aqp5* gene transcription [26]. It was also associated with a significant reduction in *Aqp5* mRNA and AQP5 protein levels in the human heart and erythrocyte membranes, respectively, suppression of the renin-angiotensin-aldosterone system, and reduced blood pressure increase in response to a high salt diet [26]. In addition, the *Aqp5*-1364A/C polymorphism has been associated with increased methylation of the promoter and decreased expression in neutrophil granulocytes from human septic patients [27]. Furthermore, strong associations have been found between this *Aqp5* polymorphism and increased 30-day survival in severe sepsis [28], reduced susceptibility for major adverse kidney events in sepsis [29], attenuated pulmonary inflammation and higher 30-day survival in acute respiratory distress syndrome [30], decreased prevalence and better resolution of acute kidney injury in patients suffering from pneumonia evoked acute respiratory distress syndrome [31], increased risk of cytomegalovirus infection after kidney transplantation [32], and less severe peritumoral brain edema in patients suffering from meningioma [33]. 

Other studies have identified some *Aqp5* mutations (p.Trp35Ser, p.Ala38Glu, p.Ile45Ser, p.Asn123Asp, p.Asn123Tyr, p.Ile177Phe, and p.Arg188Cys) causing autosomal-dominant diffuse nonepidermolytic palmoplantar keratoderma despite retaining normal expression and the ability to traffic to the cell membrane [34,35,36]. Protein modeling has suggested these mutations may provide AQP5 a gain-of-function impacting its water permeability [34]. Indeed, the substitution of amino acids lining the extracellular end of the water channel (Ile45, Ile177, and Arg188) in a highly-conserved position in extracellular loop A (Trp35) or located on the extracellular surface of AQP5 (Ala38, Asn123) may have a direct or indirect impact on water conductance [34,37]. Furthermore, molecular dynamics on mutated AQP5 (p.Arg188Cys) revealed a remarkable increase in the pore radius inside the selectivity region, facilitating the passage of water molecules and higher water permeability [38]. While it remains unclear how defective epidermal-water-barrier function associated with mutant AQP5 leads to palmoplantar hyperkeratosis, the gain of function of mutated AQP5 may accelerate keratinocyte water uptake and explain the swelling of the epidermal stratum corneum after water exposure.

A heterozygous missense mutation of *Aqp5* (p.Leu51Pro) induced autosomal dominant congenital cataracts leading to blindness in infants [39]. Transfection of the mutated *Aqp5* in human lens epithelial cells decreased AQP5 expression and increased vimentin expression through the downregulation of miR-124-3p.1 [39]. *Aqp5* knockout mice showed that a reduction in lens transparency resulted from increased vimentin expression concomitant to miR-124-3p.1 downregulation and this phenomenon was reverted using miR-124-3p.1 agomir [39]. Altogether, the data suggested AQP5 is involved in lens transparency by upregulating miR-124-3p.1 that targeted vimentin mRNA. Furthermore, it was shown that the permeability of AQP5 to water and carbon dioxide depended on the presence of Leu51 (its side chain facing the tetramer’s centra pore). Indeed, Leu51Arg mutation resulted in a permeability loss to water and carbon dioxide and converted the protein into an anion channel [40]. Further studies are necessary to assess whether the missense mutation of *Aqp5* causing autosomal dominant congenital cataract (p.Leu51Pro) would behave similarly to the Leu51Arg mutation.

A naturally occurring *rAqp5* point mutation (p.G308A) caused a reduction in saliva secretion due to impaired AQP5 trafficking and increased lysosomal degradation concomitant to possible protein conformational modification without affecting the water permeability of the channel [41]. In addition, the higher activation energy of water transport of the mutated AQP5 may prevent the transient swelling of the acini, delay the agonist-induced shrinkage, and decrease the initial transient secretion of saliva [42]. By comparing data from *Aqp5*-G103D mutant and wild-type rats, AQP5 was proposed to be responsible for the rapid agonist-induced salivary gland acinar cell shrinkage and to contribute not only to the initial phase of secretion but also to the maintenance of the necessary acinar cell volume during the steady secretion of saliva [42]. It remains to be assessed if a similar mutation occurs in humans and is associated with phenotypic alteration of saliva gland function.

## 3. Transcriptional Regulation of *Aqp5* Expression

Transcriptional regulation of *Aqp5* expression is modulated by hormonal, neuronal, and inflammatory stimuli, binding of transcription factors to gene promoter and epigenetic modifications.

### 3.1. Hormonal, Neuronal, and Inflammatory Stimuli

#### 3.1.1. Steroid Hormones

Steroid hormones have been shown to modulate *Aqp5* expression. Indeed, progesterone increased *Aqp5* mRNA expression in porcine endometrial luminal epithelial cells during the follicular phase of the estrous cycle, while estradiol had the inverse effect during the early luteal phase of the estrous cycle [43]. In mouse uterus, the *Aqp5* mRNA level was increased by estradiol but not by progesterone, suggesting abnormal *Aqp5* expression during pregnancy may participate in disrupting normal myometrial activity and reducing implantation events [44,45]. In mouse mammary gland cells, estrogen increases *Aqp5* mRNA levels, while prolactin and dexamethasone add the opposite effects [46]. 

The testosterone-induced upregulation of *Aqp5* mRNA levels in ovariectomized rats may account for the reduced uterine fluid volume measured in response to the hormone [47]. The oxytocin-induced *Aqp5* expression in the myometrium of late-pregnant rat uterus may participate in events leading to parturition in the rat [48]. In addition, the sudden increase of blood *Aqp5* levels on the last day of pregnancy may serve as an indicative marker of delivery initiation [48].

In porcine uterine tissue, *Aqp5* mRNA levels were increased by estradiol and oxytocin but decreased by progesterone [49]. In addition, growth hormone decreased *Aqp5* mRNA expression in granulosa cells from medium swine ovarian follicles, while prolactin induced an opposite effect in similar cells from large follicles. In addition, both growth hormone and prolactin increased *Aqp5* mRNA levels in theca cells from large ovarian follicles. When granulosa and theca cells from medium and large follicles were co-cultured, *Aqp5* mRNA levels were increased by the luteinizing hormone, prolactin, and growth hormone in granulosa cells from large follicles, by luteinizing hormone in theca cells from medium and large follicles and by prolactin in theca cells from medium follicles [50]. It was proposed that AQP5 may play important roles in maintaining water homeostasis of the ovarian cells of the pig involved in folliculogenesis, ovulation, and corpus luteum formation and maintenance.

In human endometrial stromal cells, the estradiol-induced increase in *Aqp5* expression consequent to the activation of the PI3K/AKT pathway enhanced cell invasion and proliferation and formation of ectopic implants in vivo [51]. Overall, these data suggest the inhibition of *Aqp5* expression and AQP5 function may provide new therapeutic options for treating endometriosis.

The presence of an estrogen-response element (ERE), identified within the *hAqp5* gene using luciferase reporter assay and electrophoretic mobility shift assay (EMSA), accounts for the estradiol-induced transcriptional regulation of *Aqp5* [51].

Corticosteroids upregulated *Aqp5* mRNA and AQP5 protein expression in adenocarcinomic human alveolar basal epithelial cells (A549) and may therefore improve pulmonary diseases with airway hypersecretion though this mechanism [52]. 

#### 3.1.2. Thyroid and Parathyroid Hormones

Thyroid hormones likely modulated AQP5 expression in the lungs, as hypothyroidism had a lowering effect while hyperthyroidism had an opposite effect [53]. These data suggest circulating thyroid hormone levels may affect lung fluid balance through several mechanisms, including the regulation of AQP5 expression and paracellular permeability. A study carried out in parathyroid hormone/parathyroid hormone-related peptide type I receptor null mice suggested parathyroid hormone may promote *Aqp5* mRNA expression in lungs [54]. As the receptors to the thyroid and parathyroid hormones belong to the family of G protein-coupled receptors linked to Gαs and Gαp, respectively, it is likely that cAMP/PKA/CREB and intracellular calcium pathways contributed to the regulation of *Aqp5* transcription. 

#### 3.1.3. Neurotransmitters

In a mouse model of primary focal hyperhidrosis, increased expression of acetylcholine and cholinergic receptor nicotinic alpha 1 subunit led to increased *Aqp5* mRNA expression [55]. Furthermore, cisatracurium, an inhibitor of acetylcholine and cholinergic receptor nicotinic alpha 1 subunit, decreased *Aqp5* mRNA and AQP5 protein expression [56]. These data suggested that increased acetylcholine levels may account for increased *Aqp5* mRNA and AQP5 protein expression leading to primary focal hyperhidrosis in patients [57]. Methacholine, a muscarinic agonist, decreased AQP5 protein expression in human nasal epithelial cells through a mechanism involving the cAMP Response Element-binding protein (CREB) inactivation and NFkB activation, suggesting muscarinic receptor stimulation may regulate *Aqp5* transcription though such signaling pathway [58]. 

β-adrenergic receptor agonists increased *Aqp5* transcription in salivary glands and lungs in vivo [59] and murine lung epithelial cell line (MLE-12) in vitro [60], likely via cAMP-dependent and CREB phosphorylation mechanisms. 

Gamma-amino butyric acid may be involved in the regulation of AQP5 expression. Indeed, gabapentin, a structural analog of gamma-amino butyric acid, increased the AQP5 expression in corneal cells and lacrimal glands through stimulation of the autonomic nervous system and direct activation of intracellular signaling cascades, including the PKA/CREB pathway [61]. These mechanisms could contribute to restoring the tear film in dry eye disease, thus reducing friction on the ocular surface and consequent ocular pain. 

Histamine downregulated *Aqp5* mRNA levels in human nasal epithelial cells by activating NFkB [62] and inhibiting CREB phosphorylation [63]. 

#### 3.1.4. Inflammatory Stimuli

Overexpression of activin, a receptor type 1, a receptor of TGFβ, led to increased *Aqp5* mRNA and AQP5 protein levels in patients with primary focal hyperhidrosis [64], suggesting that TGFβ may regulate *Aqp5* transcription. In A549 cells (alveolar type II cells) [65], primary culture of rat alveolar epithelial cells and rat lung epithelial-T-antigen negative cells (RLE-6TN; alveolar type II cells) [66], TGFβ1-induced epithelial-mesenchymal transition (EMT) was accompanied by a downregulation of AQP5 protein expression. However, the effect of TGFβ1 on *Aqp5* transcription remains to be assessed. Further studies are necessary to determine if SMAD transcription factors, involved in the TGFβ1 signaling pathway, may regulate *Aqp5* transcription.

TNFα decreased *Aqp5* mRNA and AQP5 protein expression in immortalized human salivary gland acinar cells (NS-SV-AC) [67], MLE-12 cells [68], and human bronchial epithelial cells (BEAS-2B) [69], and AQP5 protein level in conditional transgenic TNFα mice [70]. Lipoxins, endogenous lipids that mediate the resolution of inflammation, reversed the effect of TNFα by promoting *Aqp5* expression in the lung in a mouse model of acute pancreatitis-induced acute lung injury [71]. The regulation of *Aqp5* transcription by TNFα may likely involve NFkB.

IFNα upregulated both *Aqp5* mRNA and AQP5 protein expression in the primary culture of human parotid gland acinar cells [72]. Interleukin-13 (IL-13), a central regulator of Th2-dominated respiratory disorders such as asthma, abolished the increase in *Aqp5* mRNA level during airway epithelial cell differentiation [73]. In addition, reduced *Aqp5* mRNA and AQP5 protein levels were detected in lungs from an interleukin-13-induced mouse model of asthma [74]. Additional experiments need to be performed to assess the possible role of STAT transcription factors, activated upon IFNα and IL13 stimulation, in *Aqp5* transcription.

High-mobility group box 1 (HMGB1), a highly conserved nuclear protein of the alarmin family acting as a chromatin-binding factor, promotes access to transcriptional protein assemblies on specific DNA targets [75,76]. HMGB1 has been involved in Sjögren’s syndrome pathogenesis [77]. Moreover, HMGB1 decreased *Aqp5* mRNA levels in a mouse model of Sjögren’s syndrome through the activation of the toll-like receptor 4 (TLR4)/NFkB pathway [78]. In a mouse model of Sjögren’s syndrome, HMGB1 inhibition restored saliva and tear secretion [78,79], suggesting HMGB1 inhibition may represent a new therapeutic strategy for treating the disease.

### 3.2. Transcription Factors

Several transcription factors have been identified as regulators of *Aqp5* transcription during physiological and pathological processes (Figure 1).

#### 3.2.1. CREB

The cAMP-induced *Aqp5* transcription detected in several cell types [59,60,80,81] was likely linked to the presence of putative cAMP response element binding sites within the *Aqp5* gene [59]. Indeed, several studies have suggested a role for p-CREB in the cAMP-induced *Aqp5* transcription [62,82,83]. 

#### 3.2.2. NFkB

Nuclear factor kappa light chain enhancer of activated B cells (NFkB) played a role in either stimulation or inhibition of *Aqp5* transcription. Indeed, NFkB was involved in the hyperosmolar-induced increase in *Aqp5* mRNA levels in retinal pigmented epithelial cells [84]. By contrast, NFkB was suggested to inhibit the p-CREB-induced *Aqp5* transcription [62,83]. Upon lipopolysaccharide stimulation, NFkB was directly involved in the decreased *Aqp5* mRNA levels, and the effect was potentiated by interaction with p-c-Jun/c-Fos [85]. NFkB was also involved in the inhibition of *Aqp5* transcription induced by tumor necrosis factor α in MLE-12 cells [68].

#### 3.2.3. HIF1α

Hypoxia-inducible factor-1 (HIF1α) has been shown to transactivate *Aqp5* gene promoter in various cell types [86,87,88]. In mouse lung epithelial (MLE-15) cells, hyperosmolar induction of *Aqp5* mRNA expression involved the activation of ERK and mitogen-activated extracellular regulated kinase (MEK) [89]. These signaling pathways, which lead to the activation of hypoxia-inducible factor-1α (HIF1α) [90], may be involved in the hyperosmolar-induced activation of HIF1α and its binding to HIF1α binding sites to the proximal *Aqp5* gene promoter to induce transcription [86]. Decreased HIF1α activity during intervertebral disc degeneration suppressed *Aqp5* mRNA and protein expression, compromising their ability to respond to extracellular osmolarity changes [86]. However, hypoxia failed to promote *Aqp5* mRNA expression in intervertebral discs [86]. HIF1α may also increase *Aqp5* transcription in salivary glands from heat-acclimated rats [91]. 

#### 3.2.4. NFAT5

Nuclear factor of activated T cells 5 (NFAT5) upregulated *Aqp5* mRNA levels in chronic lymphocytic leukemia B cells (MEC-1 cells) and B cells isolated from patients with chronic lymphocytic leukemia [92], lung adenocarcinoma cells [93]. Furthermore, NFAT5 was involved in the hyperosmolar induction of *Aqp5* mRNA expression in human retinal pigmented epithelial cells [84], human corneal epithelial cells [94], and mouse intervertebral discs [95]. The transcriptional regulation of *Aqp5* by NFAT5 likely plays a role in cell volume regulation upon osmolar stress.

#### 3.2.5. FoxO1

Forkhead box O1 (FoxO1) stimulated the *Aqp5* gene promoter in rat salivary gland cells, and injection of FoxO1 inhibitor into mice reduced *Aqp5* expression and decreased saliva secretion [96]. Simultaneous downregulation of FoxO1 and *Aqp5* mRNA levels in patients with Sjögren’s syndrome may contribute to hyposalivation [96]. In astrocytes and microglia cells, FoxO1 was involved in the lipopolysaccharide-induced *Aqp5* expression [97]. 

#### 3.2.6. NMP4 and p300

Nuclear matrix protein 4 (NMP4), also named Zinc Finger Protein 384 (ZNF384), bound to the *Aqp5* gene promoter and induced its transactivation when overexpressed in immortalized human embryonic kidney (HEK293) cells [98]. Interestingly, upon mechanical stimulation of osteoblasts cells, the triggered signals upstream (ERK, Akt, and GSK-3β activation) and downstream (cyclin D1) of β-catenin, as well as the cytoskeleton reorganization and formation of focal adhesions between cells and extracellular substrate were enhanced upon NMP4 deficiency [99]. Therefore, NMP4 may play an inhibitory regulatory role in the response of cells to mechanical stimulation. In addition, NMP4 was shown to interact directly or indirectly with focal adhesion-associated protein p130^cas^ (CAS) [100,101], which can bind to focal adhesion-associated tyrosine kinase (FAK). Besides, AQP5 interacted with β-catenin and other junctional proteins, which displayed lower expression at the junction upon AQP5 overexpression [102]. Besides, AQP5 was not recruited to the junctions upon mechanical stimulation [102]. It was shown that co-activator p300/β-catenin interactions were necessary for p300 transcriptional activation of *Aqp5* gene in lung alveolar epithelial cell type I [103]. Further studies will be necessary to determine if another coactivator of β-catenin, such as CREB and other components of the basal transcription machinery, is also involved in such transcriptional control. Further studies will be required to precise the role of the transcriptional regulation of *Aqp5* by NMP4 and p300 in the context of cell adhesion. 

#### 3.2.7. BMAL1 and CLOCK

A relationship was established between clock genes and *Aqp5* expression in rat whole salivary glands as well as isolated acinar and ductal cells. Human, rat, and mouse *Aqp5* gene promoter analysis revealed the presence of an enhancer box (E-box) binding sequence [104,105] that acts as a protein-binding site and is necessary for the binding of Basic Helix-Loop-Helix ARNT Like 1 (BMAL1)-circadian locomotor output cycles kaput protein (CLOCK) heterodimer to the promoter region [105]. In addition, *Aqp5* mRNA expression had a rhythmic pattern similar to the circadian rhythmic expression of clock genes [106]. Therefore, *Aqp5* gene may be a clock-controlled gene and a target for the transcription factors BMAL1-CLOCK heterodimer in salivary glands. Further studies will be valuable to determine if such regulation may play a role in the physiological nocturnal salivary water secretion. 

#### 3.2.8. GATA-6

During embryonic lung development in the mouse, GATA Binding Protein 6 (GATA-6) has been shown to transactivate the *Aqp5* gene promoter and to increase the *Aqp5* mRNA expression in alveolar epithelial type 1 cells [107]. In addition, Sp1 bound to an enhancer and interacted with GATA-6 to promote epigenetic modification of *Aqp5* gene [108,109] (see Section 3.3.2). 

#### 3.2.9. *Dec1* and Achaete-Scute Family BHLH Transcription Factor 1

The differentiated embryo chondrocyte-expressed gene 1 (*Dec1*) decreased *Aqp5* mRNA expression in salivary glands from aged mice [110].

Achaete-Scute Family BHLH Transcription Factor 1 (ASCL1) increased *Aqp5* expression in mice infected with *Helicobacter pylori* [111].

### 3.3. Epigenetic Modifications

Some epigenetic modifications have been implicated in the regulation of *Aqp5* transcription, including DNA methylation, histone modifications, and micro-RNAs (miRNAs) (Figure 2). 

#### 3.3.1. DNA Methylation

Considering the high expression of *Aqp5* in several cancers (such as breast and lung), it was hypothesized that gene promoter methylation status might account for such observation. Overall, studies suggest *Aqp5* gene promoter methylation may be cell and tissue specific. 

While *Aqp5* mRNA expression was independent of its promoter methylation status in non-tumorigenic mouse mammary gland cells EpH4 [112], studies using the most recent technologies to detect gene promoter methylation using genomic DNA from tumor cells arising from cancer patients showed divergent outcomes. Age-related DNA methylation of the *Aqp5* gene promoter was inversely correlated with gene expression. In addition, *Aqp5* gene promoter methylation varies according to the type of tumor, with lower methylation and higher gene expression in basal tumors as compared to luminal tumors [113].

In human pancreatic adenocarcinoma, hypomethylation of the *Aqp5* gene promoter accounted for its high expression associated with infiltrating immune cells, suggesting *Aqp5* may represent a novel prognostic biomarker and a new immune-associated therapeutic target for patients suffering from such cancer [114]. 

Recent studies in neutrophil granulocytes from human septic patients revealed that the C-allele of the *Aqp5*-1364A/C polymorphism was associated with an increase in gene promoter methylation and decreased expression [27]. The current lack of therapeutic options to modulate single gene promoter methylation limits the therapeutic approach. Nevertheless, unspecific therapeutic intervention, such as the use of a demethylation agent (5-azacytidine (5-AZA)), has been shown to reduce endotoxemia-induced mouse inflammatory lung injury [115]. The methylation status of the *Aqp5* promoter cytosine site (nt-937) was linked to both NFkB binding and *Aqp5* expression and may be prognostically relevant in sepsis as higher methylation is associated with sepsis non-survivors [116].

High methylation of a putative CpG island of the *Aqp5* gene promoter was associated with low *Aqp5* expression in NIH-3T3 mouse fibroblasts and freshly isolated rat alveolar epithelial cells [104]. Conversely, hypomethylation of the *Aqp5* gene promoter was associated with high *Aqp5* expression in mouse MLE-12 lung epithelial cells [104]. In addition, 5-AZA treatment induced *Aqp5* expression in NIH-3T3 cells, and in vitro methylation of the *Aqp5* gene promoter inhibited the transcription of a reporter gene in mouse lung epithelial (MLE-12) cells [104]. Therefore, DNA methylation may play a role in the cell-specific expression of *Aqp5* [104]. Finally, chromatin immunoprecipitation assays revealed that endogenous Sp1 bound to a hypomethylated, but not highly methylated, *Aqp5* gene promoter. Besides, in an immortalized normal human salivary gland ductal cell (NS-SV-DC) line lacking the expression of *Aqp5*, 5-AZA induced *Aqp5* expression that led to functional water permeability [117]. In addition, untreated NS-SV-DC cells presented hypermethylation within the consensus Sp1-binding sites located within the *Aqp5* gene promoter, whereas 5-AZA-treated NS-SV-AC cells presented demethylation at CGs located within or outside consensus Sp1 binding sites that cooperatively functioned to induce *Aqp5* expression [117]. In response to retinoic acid, Sp1 was also shown to transactivate m*Aqp5* gene promoter in MLE-12 cells, leading to increased plasma membrane water permeability [118]. In murine aging model C57BL/6CrSlc mice characterized by xerostomia, 5-AZA treatment restored salivary gland function, consequently to increased *Aqp5* expression resulting from *Aqp5* gene promoter methylation [119]. These data confirm a direct link between the *Aqp5* gene promoter methylation status and its expression and further suggest that Sp1 binding to the hypo-methylated promoter likely plays a role in cell type-specific expression of the gene. Furthermore, procaine, an anticancer agent specifically inhibiting DNA methyltransferase (DNMT) 1 activity [120], stimulates *Aqp5* mRNA expression in NS-SV-DC cells by inducing CpG island demethylation at Sp1-2 and Sp1-3 sites of the h*Aqp5* gene promoter [121]. 

Polycomb Repressive Complex 2 (PRC2), involved in transcription repression, can affect the epigenetic landscape through several mechanisms, including the recruitment of DNA methyltransferases. PRC2 was shown to promote the hypermethylation *Aqp5* gene promoter in follicular lymphoma compared to benign follicular hyperplasia [122]. These data suggest the methylation status of the *Aqp5* gene promoter may be involved in lymphoma cell biology.

During embryonic salivary gland development, decreased expression of ten-eleven translocation enzyme 2 (TET2; involved in DNA demethylation through the oxidization of 5-methylcytosine (5mC) to 5-hydroxymethylcytosine (5hmC)) was observed concomitantly with increased 5hmC levels and *Aqp5* expression at embryonic day 15 (E15) [123]. Therefore, the downregulation of TET2 expression may be a critical epigenetic event that contributes to salivary gland function.

#### 3.3.2. Histone Modifications

In NS-SV-AC cells, TNFα decreased *Aqp5* mRNA expression, consequently in an epigenetic mechanism involving the suppression of histone H4 acetylation [67].

In mice, GATA-6 stimulated *Aqp5* transcription not directly through DNA binding of GATA-6 but indirectly through interactions with Sp1 that binds to an enhancer that does not contain GATA-6 binding sites [108]. In addition, GATA6/Sp1 interaction interfered with histone deacetylase 3 (HDAC3)/Sp1 binding and recruited transcriptional co-activator/histone acetyltransferase p300, leading to histone acetylation [109]. All these data suggest the GATA-6/SP1 interaction plays a role in the epigenetic modification of the *Aqp5* gene promoter during alveolar epithelial cell differentiation. 

Jumonji domain containing-3 (Jmjd3), a key histone demethylase involved in the regulation of gene expression, is involved in the *Aqp5* gene regulation and plays a role in lung development and function [124]. Indeed, changes in *Aqp5* gene expression were associated with locus-specific methylation alterations of histone 3 (H3K27 and H3K4) [124]. 

Histone H3K79 methyltransferase Dot1a repressed the mouse *Aqp5* gene promoter by binding to the promoter region comprised between −389 and −139 [125]. In the kidney, AQP5 acts as a negative regulator of aquaporin-2 (AQP2) trafficking, thereby impairing renal water reabsorption and inducing polyuria [125]. As the gene encoding Dot1a is downregulated by aldosterone, Dot1a may link excessive aldosterone to polyuria by decreasing AQP2 [126] and increasing *Aqp5* expression. Further experiments are required to uncover the physiological role of *Aqp5* gene regulation by Dot1a in other tissues. 

#### 3.3.3. miRNA

In aged mice, an upregulation of the transcription factor *Dec1* (differentiated embryo chondrocyte-expressed gene 1) decreased *Aqp5* mRNA expression [110]. The modified expression of *Dec1* and *Aqp5* mRNA levels may result from the downregulation of miR-181c-5p, miR-141-3p, and miR-374c-5p that can bind to *Dec1* mRNA and the upregulation of miR-466i-3p that may bind to *Aqp5* mRNA [110]. 

In lipopolysaccharide-induced rat lung damage of disseminated intravascular coagulation, miR-96 and miR-330 targeted *Aqp5* mRNA [127], and concomitantly decreased *Aqp5* expression was linked to pulmonary edema [128]. These data suggested that in patients with disseminated intravascular coagulation characterized among others by lung damage often consisting in pulmonary edema, miRNA may offer a new therapeutic intervention.

In hepatitis B- virus (HBV)-induced hepatocarcinoma, miR-1271-5p has been shown to target *Aqp5* mRNA, and its overexpression resulted in decreased *Aqp5* expression to block the cancer progression by reducing cell viability, migration, and invasion [129]. In similar cancers, miR-325-3p also targeted *Aqp5* mRNA, and its overexpression inhibited cell proliferation and induced cell apoptosis. 

In colorectal cancer and colorectal cancer resistant to 5-fluorouracil (a chemotherapy drug), the upregulated *Aqp5* mRNA expression was due to low expression of miR-185-3p that targets *Aqp5* mRNA [130]. Furthermore, miR-185-3p mimic enhanced the chemosensitivity of colorectal cancer cells via an effect on the *Aqp5* mRNA [130], suggesting such a therapeutic approach may be beneficial to treat 5-fluorouracil-resistant colorectal cancers.

In human breast cancer MDA-MB-231 cells, exosome-mediated delivery of miR-1226-3p, miR-19a-3p, and miR-19b-3p targeted *Aqp5* mRNA and impeded cell migration [131].

The overexpression of miR-551b-5p in human umbilical vein endothelial cells (HUVECs) promoted the expression of *Aqp5* and cell permeability [132]. As this is unlikely due to a direct effect of the miRNA on the *Aqp5* mRNA, further studies are required to address the underlying mechanisms.

Considering several miRNAs can target the *Aqp5* gene, they may be useful for the treatment of various cancers, providing the development of suitable delivery methods and miRNA mimetics. 

## 4. Post-Translational Regulation of AQP5 

Post-translational modifications have been shown to regulate the cellular function of AQPs. Phosphorylation, lipidation, glycosylation, and ubiquitination have been shown to regulate AQP5. 

### 4.1. Phosphorylation

Multiple putative protein kinase G (PKG) phosphorylation sites are present within AQP5. However, it remains to be assessed whether PKG can phosphorylate AQP5 and account for its acetylcholine-induced trafficking involving nitric oxide / cyclic 3′,5′ monophosphate (cGMP) signaling pathway causing intracellular calcium concentration increase in rat parotid acinar cells [133]. PKC may also be involved in hypoosmotic-induced AQP5 trafficking [134]. 

Two putative protein kinase A (PKA) phosphorylations sites containing Ser 156 and Thr259 may be involved in cAMP-induced AQP5 trafficking (see Section 5.2 for more details). 

### 4.2. Lipidation

Mass spectrometry analysis conducted on bovine and human lenses has revealed the palmitoylation of AQP5 on C6 located in the N-terminus end, a region highly conserved among many species (including human, bovine, mouse, rat, and pig) [135]. Furthermore, AQP5 palmitoylation seems to occur in a narrow region of the inner lens cortex but not in the lens epithelium, outer cortex, or nucleus [135]. Considering protein palmitoylation promotes lipid raft association [136], this modification may account for the presence of AQP5 in lipid raft from the bovine lens [137] and rat parotid gland [138]. However, additional experiments are necessary to determine if this post-translational modification occurs in other cell types and promotes AQP5 trafficking. 

### 4.3. Glycosylation

AQP5, which contains two putative N-glycosylation sites (Asn124, Asn125), has been suggested to be glycosylated in human cornea (based on the presence of two immunoreactive bands detected by Western blot analysis) [139] but not in human lens fibers [140]. However, further studies are necessary to confirm such post-translational modification before investigating its possible effect on AQP5. 

### 4.4. Ubiquitination

In submandibular glands from non-obese diabetic (NOD) mice with severe combined immunodeficiency disease (SCID) lacking the gene encoding the transcription factor E2F1 (NOD/SCID.E2F1*^−/−^*), AQP5 was ubiquitinated, with an increment of 24kDa in molecular weight consistent with three molecules of ubiquitin [141]. Ubiquitination of AQP5 may account for its degradation and subsequently decreased expression observed in submandibular glands from NOD/SCID.E2F1^−/−^ mice [141]. Further studies are warranted to assess if this mechanism may be responsible for hyposalivation.

In lung epithelial cells, AQP5 protein levels were also regulated in part by ubiquitination and proteasomal degradation, and drug repurposing was able to identify a compound capable of increasing AQP5 protein levels by reducing AQP5 ubiquitination and proteasomal degradation [142].

## 5. AQP5 Trafficking

Protein trafficking is an important cellular process that governs membrane protein function. Based on the high sequence homology between AQP5 and AQP2, it has been suggested that AQP5 trafficking may be regulated very similarly to AQP2 trafficking. The numerous studies on AQP2 trafficking (involved in kidney water reabsorption) [143,144] have set a precedent for the in-depth understanding of the trafficking of other AQPs, including AQP5. Several mechanisms have been involved in triggering and directing the vesicular trafficking of AQP5 to the cell plasma membrane: signal transduction involving intracellular calcium or cyclic 3,5 adenosine mono phosphate (cAMP) increase, the interaction between AQP5 and protein partners and cytoskeleton, as well as other mechanisms which remain to be further characterized (Figure 3). 

### 5.1. Involvement of Intracellular Calcium Increase

Intracellular calcium increase has been involved in AQP5 shuttling to the plasma membrane upon activation of muscarinic M3 and α1 adrenergic receptors in rat parotid salivary glands [138,145,146,147,148,149]. The nitric oxide / cGMP signaling pathway causing intracellular calcium concentration increase has also been involved in acetylcholine-induced AQP5 trafficking in rat parotid acinar cells [133]. However, it remains to be determined whether PKG phosphorylates AQP5 and, in such case, at which position(s), considering the presence of multiple potential PKG phosphorylation sites.

Histamine-induced AQP5 trafficking in *Aqp5*-transfected cells through a calcium signaling pathway [150]. In addition, hypoosmotic stress triggered AQP5 trafficking in transfected cells through a putative PKC-dependent mechanism independent of PKA and Ser156 phosphorylation [134].

Although acute regulation of AQP5 may also take place in the plasma membrane, the plasma membrane diffusion of AQP5 was not dynamically regulated through transient calcium increase and cholesterol depletion [151]. 

### 5.2. Involvement of cAMP Increase

cAMP-PKA pathway has been implicated in AQP5 trafficking in various cell types. Indeed, AQP5 trafficked to the plasma membrane upon activation of β adrenergic receptors in rat and mouse parotid glands [145,152] and upon activation of vasoactive intestinal peptide receptors in Brunner’s glands of rat duodenum [153,154]. 

In MLE-12 cells expressing m AQP5 endogenously, AQP5 trafficking induced by cAMP analog was blocked by PKA inhibitor H89 [60,155]. Moreover, cAMP stimulation had a biphasic effect on AQP5 trafficking in MLE-12 cells. Indeed, short-term exposure (minutes) induced AQP5 internalization through a mechanism involving PKA and lysosome-dependent degradation [155]. By contrast, long-term exposure (hours) induced AQP5 translocation to the plasma membrane through a mechanism involving PKA and AQP5 phosphorylation [155]. However, upon long-term exposure to cAMP analog (30 min to 6 h), AQP5 was internalized in *Aqp5*-transfected MDCK cells and the primary culture of mouse corneal epithelial cells [156]. 

In Madin-Darby canine kidney-II (MDCK-II) cells [157] and BEAS-2B human bronchial epithelial cells [158] transfected with an *Aqp5* construct, analogs of cAMP or β-adrenergic agonist-induced AQP5 trafficking to the plasma membrane that was inhibited using a PKA inhibitor (H-89).

It was suggested that two consensus PKA phosphorylation sites present within AQP5 might play a role in cAMP-induced trafficking: one at amino acids 152–156 (with Ser 156) within loop D and one at amino acids 224–262 (with Thr259) at the C-terminus. Divergent data exist concerning the role of Ser156 in AQP5 trafficking. Indeed, Ser156 was phosphorylated upon cAMP stimulation and involved in AQP5 trafficking in transfected BEAS-2B human bronchial epithelial cells [158]. In addition, AQP5 was phosphorylated on Ser156 in tumor cells [159]. Ectopic *Aqp5* expression promoted cell proliferation upon cAMP-induced Ser156 phosphorylation triggering AQP5 trafficking, suggesting a direct role for AQP5 in carcinogenesis [159]. While Ser156 was also shown not to be involved in AQP5 trafficking in transfected human embryonic kidney 293 (HEK-293) cells, it has been implicated in AQP5 membrane targeting [134]. Considering PKA inhibition increased the membrane abundance of wild-type AQP5 and AQP5 Ser156Ala and Ser156Glu mutants, an additional PKA-dependent mechanism independent of Ser156 phosphorylation was suggested to take part in AQP5 trafficking [134]. This mechanism may involve the phosphorylation of an AQP5-interacting protein partner or alternative PKA phosphorylation sites. By contrast, PKA inhibition increased the trafficking of rat AQP5 (rAQP5) expressed in MDCK-II cells, and various GFP-tagged rAQP5 mutants (Ser152Ala, Thr155Val, Ser156Ala within PKA consensus site) showed lower translocation than GFP-tagged wild-type rAQP5, suggesting dephosphorylation of PKA consensus promoted AQP5 trafficking [160]. Besides, several studies have ruled out the involvement of Thr259 phosphorylation in AQP5 trafficking as mutation of Thr259 to Ala259 in rAQP5 and mAQP5 had no effect on the cAMP-induced protein translocation [157,161]. Altogether, these data support the biphasic regulation of AQP5 trafficking involving distinct short- and long-term regulations [155]. Interestingly, Ser156 phosphorylation does not induce structural changes in AQP5 [134], in contrast to the initial hypothesis formulated based on the high-resolution X-ray crystal structure of the protein [3]. Although two consensus phosphorylation sites (Ser156 and Thr259) are present in cytoplasmic loop D and C-terminus of AQP5, the correlation between their phosphorylation and trafficking signals has not been fully determined [134,161]. It has been proposed that at least three independent mechanisms may account for *Aqp5* trafficking: Ser156 phosphorylation, protein kinase A activation, and extracellular osmolarity [134].

Using rAQP5/rAQP1 and rAQP5/rAQP8 chimera, the C-terminus end of rAQP5 has been shown to play a role in its trafficking to the plasma membrane [162,163]. These data have been confirmed using hAQP5 mutant lacking the C-terminal domain (Leu225 to Arg265) [164]. Moreover, additional hAQP5 mutants revealed the role of the 10 last C-terminal amino acids (Arg256 to Arg 265) and Leu262 in AQP5 trafficking [164]. 

A naturally occurring point mutation (G308A, Gly103 > Asp103 located within the third transmembrane domain) in the r*Aqp5* gene displayed normal water permeability but reduced trafficking to the plasma membrane and possible lysosomal degradation, both accounting for decreased apical acinar localization in salivary glands and reduced saliva flow [41]. Further studies are required to further decipher the molecular mechanisms accounting for the defective trafficking of the mutated AQP5. 

Acute regulation of AQP5 may occur in the plasma membrane as the plasma membrane diffusion of AQP5 was dynamically regulated by physiological stimuli acting through cAMP, PKA, and T259 phosphorylation [151]. However, additional studies are necessary to precise the underlying mechanisms and assess the contribution of AQP5 plasma membrane diffusion in the regulation of water flow during glandular secretions.

### 5.3. Interactions between AQP5 and Protein Partners

Some AQP5-interacting protein partners have been shown to coordinate and regulate its trafficking and function in mammalian secretory cells [10,102,165], including Na-K-Cl cotransporter 1 (NKCC1), anion exchanger 2 (AE2) [166], Transient Receptor Potential Cation Channel Subfamily V Member 4 (TRPV4) [167], Mucin 5AC (MUC5AC) [168], prolactin-inducible protein (PIP) [169,170], ezrin [171], and junctional proteins [102]. 

Direct or indirect interactions between NKCC1, AE2, TRPV4, and AQP5 (that may regulate cell volume) [166,167] and between junctional proteins and AQP5 (that may regulate cell-cell adhesion) [102] are unlikely involved in AQP5 trafficking. 

The formation of a complex between AQP5 and MUC5AC may ensure the maintenance of normal eye properties and adequate hydration of mucus gel [168]. 

The interactions between PIP and AQP5 C-terminus [169,170] and between ezrin Four-point-one, Ezrin, Radixin, Moesin (FERM) domain, and AQP5 C-terminus [171] may be involved in AQP5 trafficking. Indeed, PIP-AQP5 interaction was necessary for proper AQP5 localization in lacrimal glands, and loss of the interaction resulted in abnormal AQP5 localization in a mouse model of Sjögren’s syndrome [169]. Furthermore, decreased PIP and Ezrin expression in salivary gland acinar cells from patients suffering from Sjögren’s Syndrome resulted in abnormal AQP5 localization [170,171]. In addition, ezrin regulated AQP5 trafficking in MLE-12 cells [172]. Altogether, these data suggest that PIP and ezrin may regulate AQP5 trafficking by acting as adaptors between AQP5 and the actin cytoskeleton. 

### 5.4. Involvement of Cytoskeleton 

Interaction between AQP5-containing vesicles and cytoskeleton has been involved in AQP5 trafficking triggered by intracellular calcium increase. Indeed, inhibitors of microtubules (such as colchicine and vinblastine) and actin filaments (such as cytochalasin B) prevented totally or partially, respectively, calcium-induced AQP5 trafficking [173]. Further studies will be valuable in assessing the role of cytoskeleton dynamics on AQP5 shuttling. 

### 5.5. Involvement of Other Mechanisms

In bovine and other mammalian lenses, AQP5 trafficking occurs during the maturation of lens fibers [23]. Indeed, AQP5 is located within the cytoplasm of lens epithelial cells and young differentiating lens fiber, while its localization shifted to lens fiber cell plasma membranes upon cell maturation [174]. Furthermore, cytoplasmic AQP5 trafficked to the plasma membrane of the lens fiber cells via lysosome secretion [174], considered an unconventional protein secretion pathway [175,176]. In addition, both mechanical and pharmacological reduction in lens tension switched AQP5 labeling from the membrane to cytoplasm, suggesting AQP5 functionality can be dynamically regulated to modify lens water efflux [177]. Direct AQP5-protein interactions involved in these processes remain to be identified.

AQP5 translocated to the apical membrane of mouse mammary gland ductal epithelial cells immediately before parturition and upon progesterone stimulation, suggesting it may regulate milk osmolarity [46]. However, the involved mechanism remains to be further deciphered.

AQP5 was degraded by autophagy in submandibular glands from diabetic mice via a PI3K/Akt/mTOR signaling pathway [178] and from rats with chorda tympani parasympathetic denervation [179]. In the latter, AQP5 autophagy was reversed using an M3 muscarinic receptor agonist, cevimeline [180]. Autophagic degradation of AQP5 in liver cancer cells subjected to heat shock exerted anticancer effects [181]. Upon isoproterenol injection in mice, the AQP5 protein level in the parotid gland increased, then returned to its basal level through a proteolytic degradation involving the calpain system [152]. 

The Runt-related transcription factor (Runx) /core binding factor β (Cbfb) signaling pathway, involved in salivary gland development, regulated AQP5 trafficking in the acinar cells to allow proper saliva flow [182]. However, the precise mechanism remains to be assessed.

## 6. Downstream Effectors of AQP5

AQP5 has been shown to activate a wide variety of downstream effectors accounting for cellular functions distinct from water and carbon dioxide permeabilities, including cell migration, invasion, and EMT involved in cancer genesis (Figure 4). AQP5 has been shown to be ectopically expressed or overexpressed in various cancers [183,184], such as breast [185,186] and lung cancer [185,187,188], in which AQP5 expression levels correlated with poor prognosis [186,187]. In other cancers, including prostate [189,190], non-small cell lung [191], colon [192], and cervical [193] cancers, overexpression of AQP5 correlated with metastases to lymph nodes. 

### 6.1. Ras-MAPK Signaling Pathway

AQP5 has been shown to activate the Ras-MAPK signaling pathway in some cancers, and several mechanisms have been proposed to account for this activation.

Indeed, AQP5 phosphorylation on Ser156 (located in the second intracellular loop) by PKA promotes the binding of adaptor molecules (see Section 6.5), which trigger the activation of Ras/Raf-1/mitogen activated protein -kinase 1 and 2 (MEK1/2)/extracellular receptor kinase (ERK1/2) and cyclin-dependent kinases (CDK) that phosphorylated retinoblastoma protein (Rb). The latter induced the expression of genes involved in cell proliferation, migration, and invasion that may drive EMT [192,194,195,196].

The activation of the Rac family of small GTPase 1 (Rac1; another GTPase belonging to the Ras superfamily of small GTP-binding proteins) by AQP5 [185] may also activate the Ras-MAPK signaling pathway [197]. Additional experiments are warranted to confirm these findings and the molecular mechanism involved.

The activation of epidermal growth factor receptor (EGFR) by AQP5 can stimulate the Ras/MAPK signal transduction directly or indirectly (via the activation of phosphoinositide 3-kinase (PI3K)) to lead to cell proliferation and migration of lung cancer cells [188] and human glioma cell lines [198], and keratinocyte chemoattractant expression [199]. In human colorectal cancer cell lines, AQP5 was involved in vascular endothelial growth factor (VEGF) A expression, secretion, and concomitant angiogenesis [200], but the involvement of the Ras-MAPK signaling pathway remains to be assessed.

### 6.2. Wingless/Integrated (WNT)/β-Catenin Signaling Pathway

The activation of the Wingless/Integrated (WNT)/β-catenin signaling pathway may account for the involvement of AQP5 in EMT. In human colorectal cancer cell lines, *Aqp5* silencing altered EMT by reducing the expression of mesenchymal cell markers (N-cadherin, Vimentin, and Snail) and increasing the expression of epithelial cell marker (E-cadherin) [201,202]. The alteration of EMT-related markers mediated by *Aqp5* silencing was reversed by the upregulation of β-catenin [201]. In colorectal cancer, AQP5 expression activated the WNT/β-catenin, predicted poor clinical outcome [203], increased chemoresistance to 5-fluorouracil, and promoted tumor growth [204]. In such cancer, *Aqp5* silencing inhibited the WNT/β-catenin signaling, increased the chemosensitivity to chemotherapy-inducing apoptosis, and suppressed tumor growth, suggesting AQP5 could be a useful therapeutic target for such cancer [201,204]. 

Following *Helicobacter pylori* infection in mice inducing gastritis, transcription factor Achaete-Scute Family BHLH Transcription Factor 1 (ASCL1) increased AQP5 expression, which was responsible for the activation of the WNT/β-catenin signaling pathway [111].

### 6.3. p-Smad2/3 Pathway

In human colorectal cancer cell lines, AQP5 induced the activation of phosphorylated-mothers against decapentaplegic homolog *2*/3 (p-Smad2/3), which resulted in EMT [202]. These data suggested that p-Smad2/3 may form a complex with others against decapentaplegic homolog 4 (Smad4) and translocate to the nucleus to regulate target gene transcription leading to the stimulation of EMT [205]. 

Further experimentation is necessary to determine if the recruitment of proteins by AQP5 is involved in the activation of the p-Smad2/3 pathway. 

### 6.4. p-Akt Pathway

AQP5 overexpression induced the phosphorylation of AKT, resulting in mouse cornea epithelial cell proliferation promoting tumorigenesis and apoptosis [206]. However, the involved mechanisms remain to be analyzed. AKT activation by AQP5 also stimulates NFkB, which can regulate the transcription of target genes, including that of AQP5 [199]. In hepatocellular carcinoma, AQP5 promoted cell invasion and tumor metastasis in vitro and in vivo and induced EMT involving NFkB activation [207]. 

### 6.5. Recruitment of Proteins

The recruitment of proteins can regulate many cellular processes, including gene expression, cell adhesion and migration, and cell proliferation. AQP5 has been shown to recruit some proteins through the presence of protein binding motifs. Indeed, a homologous RGN motif present within the AQP5 extracellular connecting loop C (CL3) [208] has been shown to be involved in fibronectin binding [209] and thereby may play a role in cell migration. In addition, the binding of phosphorylated extracellular loop D fragment of AQP5 (amino acids 88–182) to the SH3 domains of Src protooncogene (Src), Lyn protooncogene (Lyn) and GRB2 Related Adaptor Protein 2 (Grap2) [187] has been shown to play a role in cell migration, invasion and EMT [210,211,212], possibly by activation RAS or RAC1 and activating the MAPK pathway (see Section 6.1). AQP5 is likely recruiting the proteins via a diproline peptide sequence (RTSPVGSP; amino acids 154–161) bearing sequence similarity to the SH3 binding consensus site [187]. Both phosphorylation of AQP5 on Ser156, located within the SH3 consensus binding site, and Asn 185, located within an NPA motif, have been suggested to be critical for the AQP5-induced cell migration and invasion [187]. Interestingly, the Src family of kinases, including Src and Lyn, have been involved in cell volume regulation by modulating the activity of ions transports, some of which are identified as AQP5-interacting protein partners (i.e., TRPV4; NKCC1, and NHE) [213]. Therefore, additional experiments will be valuable to assess the role of Src and Lyn on the cell volume regulation involving AQP5.

AQP5 has also been involved in the regulation of cytoskeleton dynamics through the activation of the Ras-MAPK signaling pathway inducing actin network reorganization (see Section 6.1). Moreover, in epithelial cells, AQP5 C-terminus seemed to efficiently promote microtubule stability assembly that may regulate paracellular water permeability [214,215]. Moreover, *Aqp5* knockout mice displayed decreased expression of tight junctions proteins forming the tight junction strands (including claudins and occludin) and involved in scaffolding (including zonula occludens-1 (ZO-1)) which may account for decreased paracellular water permeability [215]. Concomitant to *Aqp5* knockout, the loss of interaction between AQP5 and junctional protein [102] may also participate to decreased paracellular water permeability. 

Overall, further experiments are necessary to further decipher the molecular mechanism linking the recruitment of proteins by AQP5 to EMT, regulation of the cytoskeleton involved in cell migration and adhesion, and paracellular permeability.

## 7. Conclusions

Several mechanisms account for the dynamic upstream regulation of AQP5. Indeed, single nucleotide polymorphisms, mutations, and transcriptional control of *Aqp5* gene have been shown to impact the expression, trafficking, and/or function of the protein. Moreover, trafficking and localization of AQP5 can also be regulated by post-translational modifications, intracellular signaling pathways (calcium and cAMP-PKA), protein interacting partners, cytoskeleton interaction, and other mechanisms. Furthermore, the dynamic regulation of these upstream events will likely impact not only the AQP5 permeability to water and carbon dioxide but also other cellular pathways resulting from the activation of downstream effectors of AQP5. Currently, these downstream effectors include the Ras-MAPK signaling pathway, WNT/β-catenin signaling pathway, p-Smad2/3 pathway, and recruitment of proteins involved in a variety of biological effects. Overall, a deeper understanding of these finely tuned upstream and downstream mechanisms will improve the current comprehension of physiological and pathophysiological processes involving AQP5. 

## Figures and Tables

**Figure 1 ijms-24-01889-f001:**
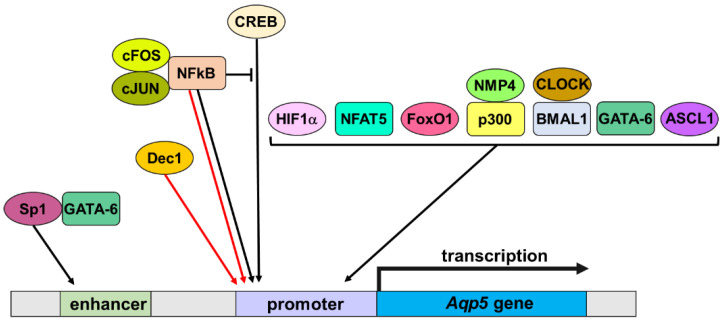
Transcription factors regulating *Aqp5* transcription. ASCL1: Achaete-Scute Family BHLH Transcription Factor 1; BMAL1: Basic Helix-Loop-Helix ARNT Like 1; CLOCK: circadian locomotor output cycles kaput protein; CREB: PKA/cAMP Response Element-binding protein; *Dec1*: differentiated embryo chondrocyte expressed gene 1; FoxO1: Forkhead box O1; GATA-6: GATA Binding Protein 6; HIF1α: hypoxia-inducible factor-1; NFAT5: nuclear factor of activated T cells 5; NFkB: Nuclear factor kappa B; NMP4: Nuclear matrix protein 4—also named Zinc Finger Protein 384 (ZNF384). Black and red arrows, respectively, indicate a positive or negative effect on *Aqp5* gene transcription. The black bar indicates an inhibitory effect.

**Figure 2 ijms-24-01889-f002:**
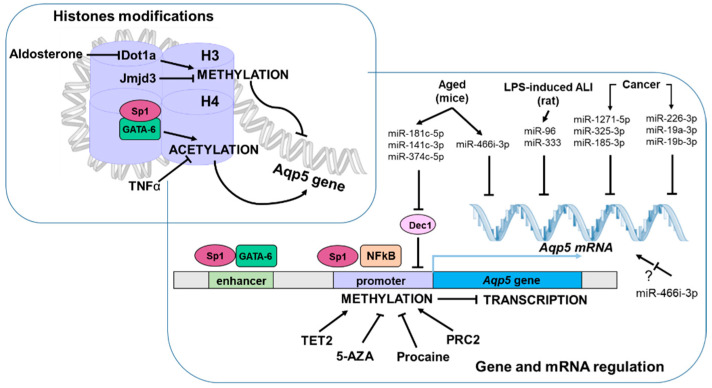
Epigenetic modifications regulating *Aqp5* transcription. Synoptic view of known epigenetic regulation of *Aqp5* expression across different cell types and medical conditions. 5-AZA: 5-Azacitidine; ALI: Acute Lung Injury; *Aqp5*: Aquaporin-5; *Dec1*: Differentiated embryo chondrocyte expressed gene 1; Dot1a: Disruptor of telomeric silencing 1a; GATA-6: GATA-binding factor 6; H3: Histone 3; H4: Histone 4; Jmjd3: Jumonji domain-containing protein-3; LPS: Lipopolysaccharides; miR: MicroRNA; mRNA: Messenger RNA; NFkB: Nuclear factor kappa-light-chain-enhancer of activated B cells; nt-937: *Aqp5* promoter methylation site; PRC2: Polycomb Repressive Complex 2; Sp1: Specificity protein 1 transcription factor; TET2: ten-eleven-translocation 2; TNFα: Tumor necrosis factor. The black bar indicates an inhibitory effect.

**Figure 3 ijms-24-01889-f003:**
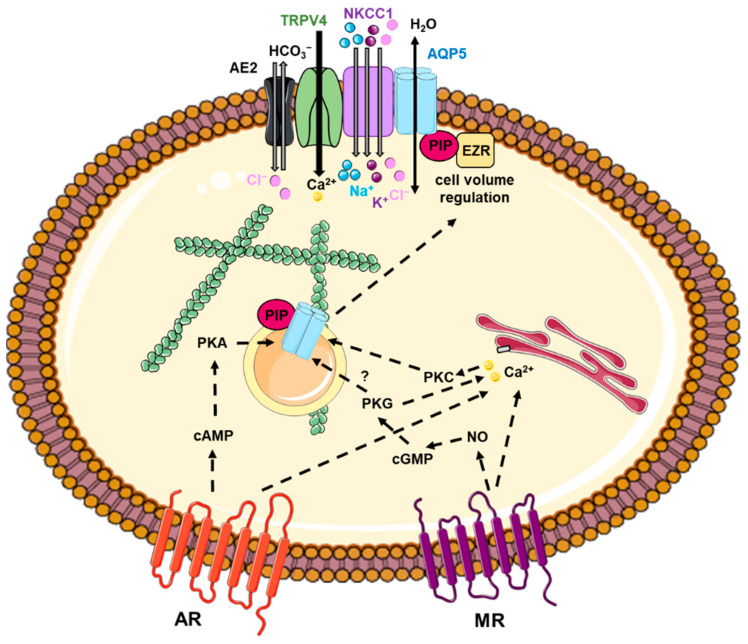
Mechanisms involved in AQP5 trafficking. Ach: acetylcholine; AE2: anion exchanger 2; AR: adrenergic receptor; cAMP: cyclic adenosine 3′,5′ monophosphate; cGMP: cyclic guanosine 3′,5′ monophosphate; EZR: ezrin; MR: muscarinic receptor; NKCC1: Na-K-Cl cotransporter 1; NO: nitric oxide; PIP: prolactin-inducible protein; PKA: protein kinase A; PKC: protein kinase C; PKG: protein kinase G; TRPV4: transient Receptor Potential Cation Channel Subfamily V Member 4.

**Figure 4 ijms-24-01889-f004:**
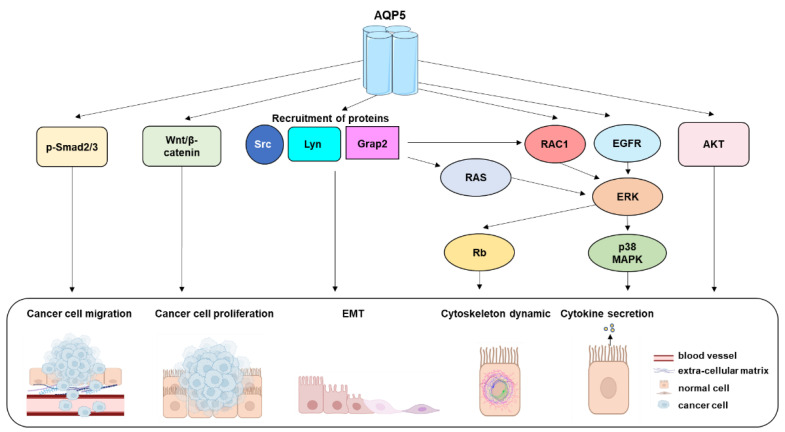
Downstream effectors of AQP5. AKT: Serine-Threonine Protein Kinase; EGFR: epidermal growth factor receptor; EMT: epithelial-mesenchymal transition; ERK1/2: extracellular receptor kinase; Grap2: GRB2 Related Adaptor Protein 2; Lyn: Lyn protooncogene; p38 MAPK: p38 mitogen-activated protein kinase; RAC1: Rac family of small GTPase 1; Ras: RAS superfamily of small GTP-binding proteins; p-Rb: phosphorylated retinoblastoma protein; p-Smad2/3: phosphorylated-mothers against decapentaplegic homolog *2*/3; Src: Src protooncogene; Wnt: Wingless/Integrated.

## Data Availability

Not applicable.

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
