# Peer review of "Aquaporin-5 Dynamic Regulation"

_ijms, 2023, doi:10.3390/ijms24031889_

Round 1

Reviewer 1 Report

The overall aim of the review to survey multiple levels involved in dynamic regulation of AQP5 is a good idea and could be a welcome contribution to the field. The Introduction in Section 1 is a well written summary of the general properties, structure and tissue specific patterns of expression of AQP5, with some interesting connections to pathological conditions. Section 2 connects genetic alterations with functional outcomes. The main concern with the MS is that the subsequent sections of the article are not equally as clear and concise in terms of presentation.

Section 3 lacks a clear plan. This section could be organized into coherent themes based on common mechanisms rather than presented as long (somewhat random) bullet lists of agents. In 3.3, signaling pathways involved in transmitter mediated regulation of AQP5 expression need to be clearly explained for acetylcholine, adrenaline, GABA, and histamine. Similarly, the links from inflammatory stimuli to transcription of AQP5 could be more clearly explained. In each case, distinguishing between correlation -vs- causal relationships in the observed outcomes is essential. The elucidation of specific mechanisms which directly regulate AQP5 transcriptional activity (as per the section heading) is not obvious for many of the signaling molecules listed.  Ideas in Section 3.4 would benefit from organization into functional themes. The  first sentence of  3.4.1 ruling out promoter methylation seems to contradict the rest of the information covered in this subsection.

Section 4 on post-translational regulation is surprisingly brief, missing a number of candidate mechanisms, and needs to be expanded. A comparison with similar mechanisms in other AQP classes could add depth and insight.

Section 5 on protein trafficking would benefit from a broader perspective on this cellular process in governing membrane protein functions, and at least mentioning AQP2 which set the precedent, before focusing down on AQP5.

Figure 1 would benefit from including outcomes and relationships between factors, to convey a more comprehensive summary of the main points.

In Fig 2, the consideration  of associated medical conditions is a laudable goal, but given finite space, this unfortunately ends up being inaccurate and overgeneralized. It seems unlikely for example that the uniform action of all cancers is to block all methylation of the AQP5 promoter.  Just an idea the authors might consider would be reducing the scope of the topic covered in each illustration (perhaps to just one medical condition?), and then generating a series of figures to tackle the different conditions of interest to capture a more complete picture.  

Minor typographic errors (e.g., missing letters; single sentence paragraphs; use of the word "incriminated" instead of "implicated") will be easily corrected.

Author Response

We thank the reviewer for his constructive comments.

The overall aim of the review to survey multiple levels involved in dynamic regulation of AQP5 is a good idea and could be a welcome contribution to the field. The Introduction in Section 1 is a well written summary of the general properties, structure and tissue specific patterns of expression of AQP5, with some interesting connections to pathological conditions. Section 2 connects genetic alterations with functional outcomes. The main concern with the MS is that the subsequent sections of the article are not equally as clear and concise in terms of presentation.

Section 3 lacks a clear plan. This section could be organized into coherent themes based on common mechanisms rather than presented as long (somewhat random) bullet lists of agents.

A brief introduction was added to section Section 3 to clarify the organization of the section. Furthermore, Section 3 was reorganized using more suitable section headings.

In 3.3, signaling pathways involved in transmitter mediated regulation of AQP5 expression need to be clearly explained for acetylcholine, adrenaline, GABA, and histamine.

As requested, signaling pathways involved in transmitter mediated regulation of AQP5 were more clearly explained for acetylcholine, adrenaline, GABA and histamine.

 Similarly, the links from inflammatory stimuli to transcription of AQP5 could be more clearly explained. In each case, distinguishing between correlation -vs- causal relationships in the observed outcomes is essential.

As requested, the links between inflammatory stimuli to transcription of AQP5 were more clearly explained.

The elucidation of specific mechanisms which directly regulate AQP5 transcriptional activity (as per the section heading) is not obvious for many of the signaling molecules listed. 

Section 3 was reorganized using more suitable section headings. Clarification of specific mechanism directly regulating AQP5 transcriptional activity were provided for each stimuli, when possible.

Ideas in Section 3.4 would benefit from organization into functional themes. The first sentence of  3.4.1 ruling out promoter methylation seems to contradict the rest of the information covered in this subsection.

The first sentences of section 3.4.1 were modified according to reviewer’s comment.

Section 4 on post-translational regulation is surprisingly brief, missing a number of candidate mechanisms, and needs to be expanded. A comparison with similar mechanisms in other AQP classes could add depth and insight.

As requested, we added a small introductive paragraph for Section 4 indicating post-translational modifications have been shown to regulate the cellular functions of AQPs, but remained focused on AQP5 (the topic of this review). In addition, Section 4 was expanded to discuss other post-translational mechanisms that have been shown to be relevant to AQP5 such as 4.1. phosphorylation (a brief summary was given under this section 4 as more details were already provided under section 5) and 4.4. ubiquitination.

Section 5 on protein trafficking would benefit from a broader perspective on this cellular process in governing membrane protein functions, and at least mentioning AQP2 which set the precedent, before focusing down on AQP5.

Beginning of Section 5 was modified according to the reviewer’s comment to explain the precedent set by the studies on AQP2 trafficking.

Figure 1 would benefit from including outcomes and relationships between factors, to convey a more comprehensive summary of the main points.

As requested, Figure 1 was modified to indicate outcomes and relationships between transcription factors to convey a more comprehensive summary of the main points.

In Fig 2, the consideration of associated medical conditions is a laudable goal, but given finite space, this unfortunately ends up being inaccurate and overgeneralized. It seems unlikely for example that the uniform action of all cancers is to block all methylation of the AQP5 promoter.  Just an idea the authors might consider would be reducing the scope of the topic covered in each illustration (perhaps to just one medical condition?), and then generating a series of figures to tackle the different conditions of interest to capture a more complete picture. 

As requested, to ovoid overgeneralization, we removed the terms sepsis and cancer for the promoter methylation status. We thank the reviewer for suggesting preparing a series of figures for each medical conditions. However, we choose not to do so to avoid multiplying figures for this section considering all information related to medical conditions were provided within the text.

 Minor typographic errors (e.g., missing letters; single sentence paragraphs; use of the word "incriminated" instead of "implicated") will be easily corrected.

As requested, minor typographic errors have been corrected.

Reviewer 2 Report

The manuscript submitted for review describes in detail the latest research on AQP5. The authors presented all the data and results regarding aquaporin 5. In my opinion, well-constructed subsections in the work that organize the existing knowledge about the physiological and pathophysiological processes in which AQP5 is involved. Information contained in the work systematize and extend the knowledge about AQP5. In my opinion the manuscript in this form can be further processed and published in Internatinonal Journal of Molecular Sciences.

Author Response

The manuscript submitted for review describes in detail the latest research on AQP5. The authors presented all the data and results regarding aquaporin 5. In my opinion, well-constructed subsections in the work that organize the existing knowledge about the physiological and pathophysiological processes in which AQP5 is involved. Information contained in the work systematize and extend the knowledge about AQP5. In my opinion the manuscript in this form can be further processed and published in Internatinonal Journal of Molecular Sciences.

We thank the reviewer for the very positive comments on the review paper.

Reviewer 3 Report

This is a very comprehensive and complete review. The writing is very good and clear.

Are the figures the original work of the authors? They are very well done and should be attributed to the author(s)

1. What is the main question addressed by the research?
 Review of the literature and knowledge base for Aquaphorin 5
2. Do you consider the topic original or relevant in the field?
YES
3. Does it address a specific gap in the field?
This is a review. It is not original but serves a specific gap in the field as there is no present comprehensive review of this topic.
3. What does it add to the subject area compared with other published material?
It puts all of this information in one place where the reader can find out and be updated on AQ 5.
4. What specific improvements should the authors consider regarding the methodology?  What further controls should be considered?
N/A this is a review not  a research paper
5. Are the conclusions consistent with the evidence and arguments presented and do they address the main question posed?
YES
6. Are the references appropriate?
YES
7. Please include any additional comments on the tables and figures. The figures are very nice and detailed.
They should be referenced as original artwork by the authors or "a derivative of someone elses"

Author Response

This is a very comprehensive and complete review. The writing is very good and clear.

Are the figures the original work of the authors? They are very well done and should be attributed to the author(s)

  1. What is the main question addressed by the research?
    Review of the literature and knowledge base for Aquaphorin 5
    2. Do you consider the topic original or relevant in the field?
    YES
    3. Does it address a specific gap in the field?
    This is a review. It is not original but serves a specific gap in the field as there is no present comprehensive review of this topic.
    3. What does it add to the subject area compared with other published material?
    It puts all of this information in one place where the reader can find out and be updated on AQ 5.
    4. What specific improvements should the authors consider regarding the methodology?  What further controls should be considered?
    N/A this is a review not  a research paper
    5. Are the conclusions consistent with the evidence and arguments presented and do they address the main question posed?
    YES
    6. Are the references appropriate?
    YES
    7. Please include any additional comments on the tables and figures. The figures are very nice and detailed.
    They should be referenced as original artwork by the authors or "a derivative of someone elses"

We thank the reviewer for the positive comments on the review paper and the quality of the figures. The figures are original and have not been adapted from other papers (otherwise, we would have mentioned “adapted from” in the text). As of common tacit rule, no comments were made in text to stipulate all presented figures are original.

Reviewer 4 Report

Dear Authors,

There are so many papers regarding focusing on regulation of AQP5; however, this reviewer thinks little is known about physiological and pathological importance. The authors well reviewed so many papers compactly. This reviewer did not find any issues in this submitted review article. This is the excellent review.

Thank you very much.

Author Response

Dear Authors,

There are so many papers regarding focusing on regulation of AQP5; however, this reviewer thinks little is known about physiological and pathological importance. The authors well reviewed so many papers compactly. This reviewer did not find any issues in this submitted review article. This is the excellent review.

Thank you very much.

We thank the reviewer for finding this review excellent.

Round 2

Reviewer 1 Report

The authors have addressed the comments raised in the prior review. I have no further concerns.